# Social and structural barriers and facilitators to HIV healthcare and harm reduction services for people experiencing syndemics in Manitoba: study protocol

Zulma Vanessa Rueda [ORCID],[1] Margaret Haworth-Brockman,[2,3] Cheryl Sobie,[1] Enrique Villacis,[1] Linda Larcombe,[4] Katharina Maier,[5] Kathleen Deering,[6,7] Julianne Sanguins,[3] Kimberly Templeton,[8,9] Lauren MacKenzie,[4,8] Laurie Ireland,[8,9] Ken Kasper,[4,8] Michael Payne,[9] Jared Bullard,[1,10] Andrea Krusi,[6,7] Neora Pick,[11] Tara Myran,[12] Adrienne Meyers,[13] Yoav Keynan[1,2,3,4]

For numbered affiliations see end of article.

**Correspondence to**
Dr Zulma Vanessa Rueda;
zulma.rueda@umanitoba.ca

## ABSTRACT

**Introduction** In Manitoba, Canada, there has been an increase in the number of people newly diagnosed with HIV and those not returning for regular HIV care. The COVID-19 pandemic resulted in increased sex and gender disparities in disease risk and mortalities, decreased harm reduction services and reduced access to healthcare. These health crises intersect with increased drug use and drug poisoning deaths, houselessness and other structural and social factors most acutely among historically underserved groups. We aim to explore the social and structural barriers and facilitators to HIV care and harm reduction services experienced by people living with HIV (PLHIV) in Manitoba.

**Methods and analysis** Our study draws on participatory action research design. Guiding the methodological design are the lived experiences of PLHIV. In-depth semi-structured face-to-face interviews and quantitative questionnaires will be conducted with two groups: (1) persons aged ≥18 years living or newly diagnosed with HIV and (2) service providers who work with PLHIV. Data collection will include sex, gender, sociodemographic information, income and housing, experiences with the criminal justice system, sexual practices, substance use practices and harm reduction access, experiences with violence and support, HIV care journey (since diagnosis until present), childhood trauma and a decision-making questionnaire. Data will be analysed intersectionally, employing grounded theory for thematic analysis, sex-based and gender-based analysis and social determinants of health and syndemic framework to understand the experiences of PLHIV in Manitoba.

**Ethics and dissemination** We received approval from the University of Manitoba Health Ethics Research Board (HS25572; H2022:218), First Nations Health and Social Secretariat of Manitoba, Nine Circles Community Health Centre, Shared Health Manitoba (SH2022:194) and 7th Street Health Access Centre. Findings will be disseminated using community-focused knowledge translation strategies identified by participants, peers, community members and organisations, and reported in conferences, peer-reviewed journals and a website (www.alltogether4ideas.org).

## STRENGTHS AND LIMITATIONS OF THIS STUDY

⇒ A diverse research advisory and peer research team, and comprehensive data collection tools ensure we accurately capture people's insights into HIV care and harm reduction services.

⇒ The intersectional lens would provide a better understanding of the current social and structural health inequalities experienced by people living with HIV.

⇒ The sample may not represent all circumstances since data collection will occur in two metropolitan areas and with individuals who can dedicate time to the study.

## INTRODUCTION

Approximately 36.9 million people live with the HIV.[1 2] Despite global and local initiatives to address HIV,[2–5] recent findings suggest a resurgence of diagnoses among North and South American and European countries.[1] In Canada, HIV diagnoses have steadily increased among women and people who inject drugs (PWID).[6 7] In Manitoba, Canada, there is an over-representation of females, PWID, Indigenous peoples and people experiencing houselessness among people newly diagnosed with HIV.[8] Heterosexual sex and injection drug use have become the most common mode of HIV transmission since 2018, and methamphetamine is the main injected drug.[8] In Manitoba, there is an emerging pattern of people living with HIV

(PLHIV) not returning for regular care and treatment. Manitoba also has not reached any of the UNAIDS' and Canada's 95-95-95 goal (ie, 95% of all PLHIV know their status, 95% of those diagnosed receive antiretroviral treatment and 95% of those on treatment achieve viral suppression).[9 10]

The care continuum for PLHIV encompasses several interconnected components (eg, diagnosis, linkage and engagement to care).[11 12] Linkage to care has been associated with positive health outcomes and reduced HIV mortality.[12 13] In a systematic qualitative review, Tso *et al*[12] found that interventions focused on increasing community participation and literacy, creating community outreach mobile teams, integrating HIV-specific services in primary care and providing substance use services improved linkage to care.[12] On the other hand, lack of information, anxiety, fear and stigma after an HIV diagnosis were barriers to linkage to care.[12] Moreover, Hall *et al*[14] found that the leading reasons for people continuing HIV care were the use of community health workers and lay health workers, incorporating technology with health, support from friends and family, access to intensive case management and positive interpersonal relationships with service providers.[14] Personal and community stigma and discrimination, fear of HIV disclosure and service provider shortages were among the leading barriers to PLHIV staying in care.[14]

Previous findings have also highlighted how HIV experiences are shaped by biological sex and gender identities.[15–18] Ostrach and Singer[18] emphasise the importance of biological, social and political factors that place women at an increased risk of acquiring HIV. The authors suggest integrating sex-based and gender-based analyses in HIV research to accurately capture the impact of intersectional identities on PLHIV.[18] Argento *et al* found that most women (ie, cis-women and trans-women) living with HIV, who participate in sex work and may or may not use methamphetamines had experienced physical or sexual abuse during childhood and adolescence, and those who initiated methamphetamines use were more likely to have experienced childhood abuse.[15] Their study emphasises the need to consider childhood trauma in understanding health and social circumstances later in life.[15] The Canadian prairie provinces (Alberta, Saskatchewan and Manitoba) report higher violence against women than the rest of Canada.[19] Findings in these provinces highlight the harm violence has on women's lives[20] and emphasise the importance of considering women's contexts and intersecting identities (eg, cultural background, gender, disability) when developing person-centred interventions.[20–23]

The COVID-19 pandemic has dramatically affected the health of many individuals, especially among people already burdened by social and structural health inequities.[24] The pandemic also exacerbated mental health symptoms in many groups.[25] Data from Manitoba show that older adolescents and young adults self-reported increased stress/anxiety and depression, alcohol consumption and substance use and conflict with family members and intimate partner,[26] with those self-identifying as women experiencing higher mental health burden,[27 28] financial hardship and interpersonal conflicts.[26] For HIV care, the reduction or complete closure of HIV treatment centres and services in some jurisdictions placed the cascade of care for PLHIV at risk of breaking down.[24] Among PWID, COVID-19 restrictions limited safe spaces for substance use, disrupted drug supplies and restricted the availability of medical, community and traditional resources.[29]

Harm reduction encompasses strategies to reduce the health, social and economic factors that harm PWID and their networks.[30] The Public Health Agency of Canada conducted an online survey among people self-identifying as PWID in the past 6 months to understand how sexually transmitted and bloodborne infections (STBBIs) and harm reduction services changed during COVID-19.[31] The survey reported 52% of respondents increased their personal use of methamphetamine, and more than half increased their use of alcohol and other drugs.[31] As well, 50% of PLHIV had challenges getting to a provider or clinic since the onset of COVID-19.[31] More than half of the respondents had difficulty getting STBBI-related services, needle and syringe distribution programmes and naloxone training.[31] While these results highlight concerning trends, data were not reported by sex and gender, hindering an intersectional analysis. Also, the study's online modality may not have reached PWID who may also be coping with unstable housing, poverty and other structural factors, likely underestimating the disruption of services. Locally, the province of Manitoba reported record fatalities from toxic drug overdoses,[32] and PWID described harm reduction services limitations due to COVID-19.[29] However, more research is needed to understand specific service limitations and how these limitations interact with other health crises and social inequities.

Taken together, these trends suggest that an intersectional lens is needed to understand how social, structural and programmatic factors affect HIV care and harm reduction services among PLHIV, including in the context of COVID-19 disruptions. This paper reports a study protocol that aims to address the question: What are the social and structural factors that women, men and non-binary PLHIV in Manitoba are experiencing that prevent or enable HIV care and harm reduction services, before and during COVID-19 pandemic?

## STUDY OBJECTIVES AND OUTCOMES

This study explores the social and structural factors that affect the access, linkage and retention in HIV healthcare and harm reduction services among women, men and non-binary persons living with HIV (linked and non-linked to care), before and during COVID-19.

It is important to note that this study is one component of a more extensive mixed-methods project aimed at understanding the epidemiology of HIV and syndemics experienced by PLHIV in Manitoba.

## Outcomes

1. To gain an in-depth understanding of the social and structural barriers and facilitators faced by women, men and non-binary persons living with HIV, affecting access, linkage and retention in HIV healthcare and access harm reduction services, before and during COVID-19 pandemic.
2. To understand how the COVID-19 pandemic has affected access, linkage and retention in HIV healthcare and harm reduction services for women, men and non-binary persons living with HIV.

## METHODS AND ANALYSIS

### Conceptual framework

The syndemic framework proposed by Singer[33] focused on understanding the interconnectedness of mutually reinforcing biomedical crises within their socioeconomic contexts. The syndemic theory 'involves the adverse interaction between diseases and health conditions of all types (eg, infections (HIV and STBBI), chronic non-communicable diseases, mental health (issues), behavioural conditions, (substance use), toxic exposure and malnutrition) and are most likely to emerge under conditions of health inequality caused by poverty, stigmatisation, stress, (trauma) or structural violence'.[33 34]

Imperative to syndemic theory is the recognition that disparate social conditions enable health crises and maintain and reinforce them.[33] Social and structural health inequities that disproportionately affect seldom-heard and marginalised populations in Manitoba require understanding of health as a social construct, rather than a biological process, as defined by the WHO Conceptual Framework for Action on the Social Determinants of Health.[35] The framework describes how social, economic and political values and policies create a socioeconomic position, for people in societies, influencing opportunities for income, education, depending on race/ethnicity, gender and other factors.[35] These structural determinants of health inequities interact with intermediary determinants of health such as material circumstances, and psychosocial, biological (including sex) and behavioural factors.[35] Using health as a social construct helps to push understanding of inequities beyond a focus on individual health and behaviours, framing health as a social construct dependent on the interplay of numerous interacting systems and circumstances.

This syndemic theory is particularly valuable in the context of communicable diseases and seldom-heard and marginalised peoples' health, as it brings interacting diseases and inequalities together, and it emphasises the political, social, historical and economic factors that have negatively affected their health outcomes.[34 36] Figure 1 summarises our approach for this study.

A comprehensive framework that includes structural and social factors is needed in Manitoba to understand the experiences of PLHIV that prevent and enable access, linkage and retention in HIV care and harm reduction services. Our methodology is intersectional by considering how sex and gender, cultural background, socioeconomic factors (including housing), trauma and violence, substance use, empowerment and decision making and other factors intersect[37] to affect individuals' experiences of HIV care and harm reduction services before and during COVID-19 pandemic.

### Patient and public involvement

This study is grounded in principles of social justice, equity and the understanding that PLHIV are experts

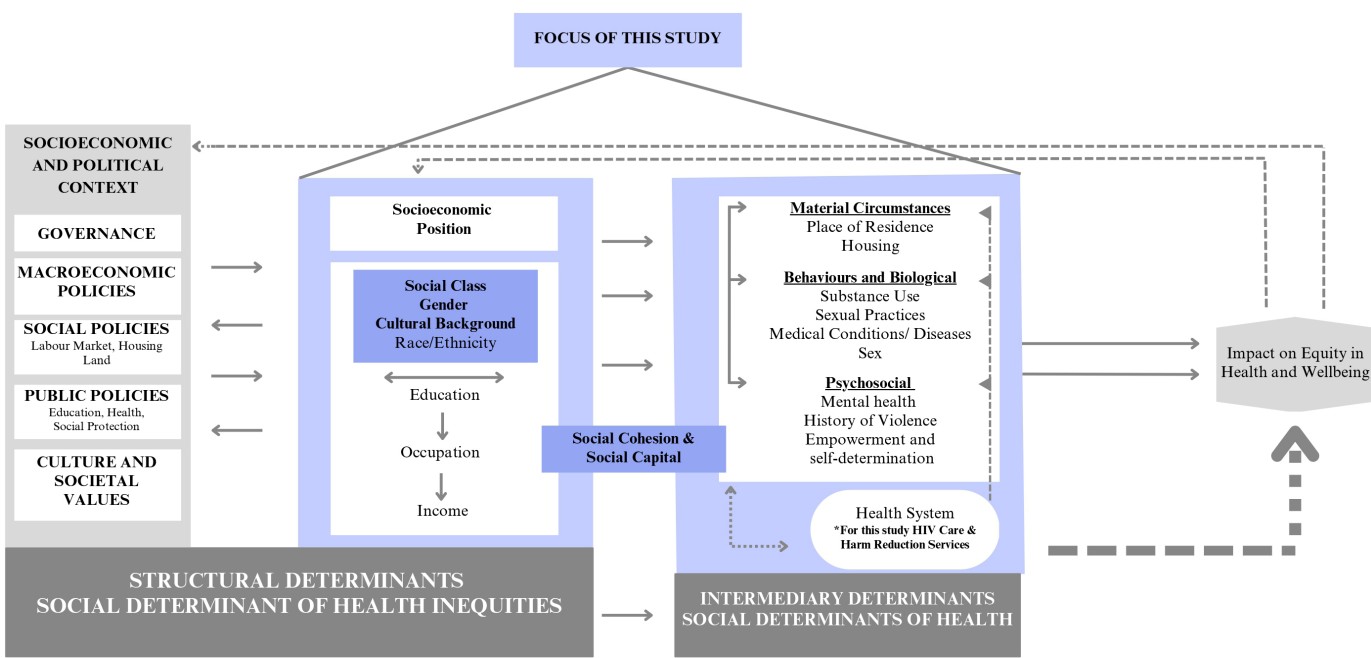

**Figure 1** Structural and social determinants[35] adapted to the main focus of our study.

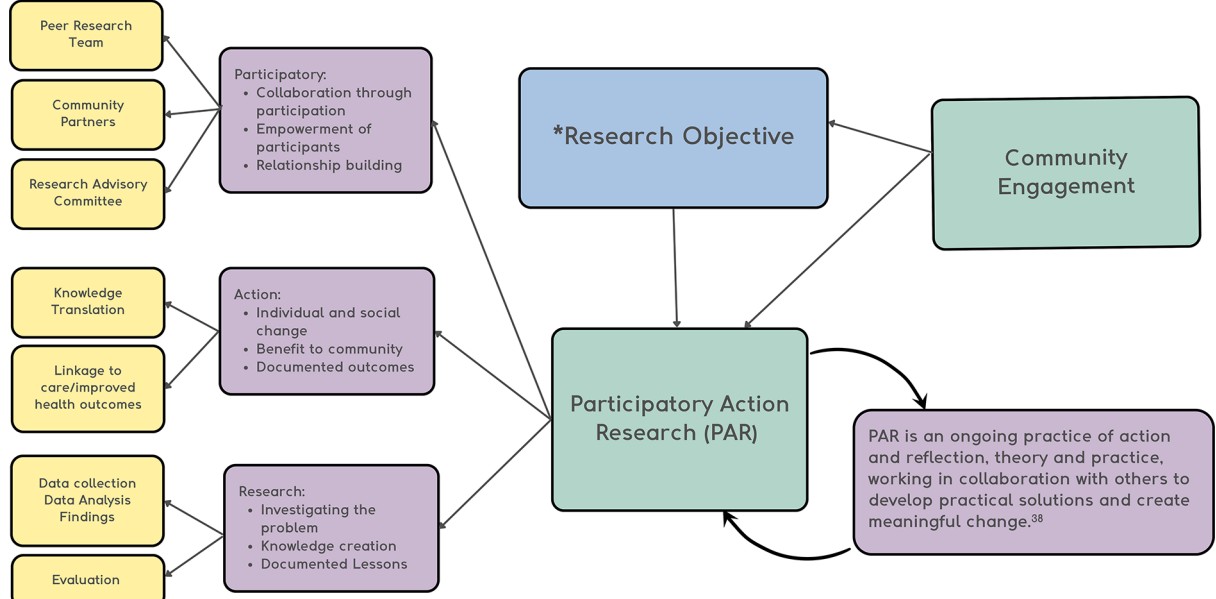

**Figure 2** Participatory engagement process. The participatory engagement process and how participatory action research (PAR will be used to achieve the outcomes related to this study.[49–51] *This study explores the social, structural and programmatic factors that affect the access, linkage and retention to HIV care and harm reduction services among women, men and non-binary persons living with HIV (linked and non-linked to care), before and during COVID-19.

whose voices should be central to the research design and process. Accurate findings are achievable, robust and meaningful only if driven by people with lived experiences. Therefore, this study uses community engagement grounded in participatory action research (figure 2).

This project is a collaborative multidisciplinary study connecting peers, community-based researchers, clinicians and academic researchers across Canada. To ensure this project remains grounded in the lived experience of PLHIV and PWID, we partnered with several community-based organisations, convened a diverse Research Advisory Committee (RAC) and created a Peer Research Team (PRT) of people with lived experience.

Before the project commencement, the principal investigators held >10 consultations with community members and peers, including PWID, PLHIV and Indigenous peers impacted by the Sixties Scoop (ie, children were systematically removed from their homes and placed into the child welfare system[38 39]). During these consultations, the community members' personal stories illustrated the complex intersections of sex, gender, stigma, substance use, colonial and structural violence and other factors affecting care. These consultations provided insights into the interconnectedness of ongoing health crises and helped focus our research questions.

Several study co-investigators are the lead facilitators of HIV care and prevention in Manitoba, including The Manitoba HIV Program, which provides information, specialised care and support to approximately 2000 PLHIV in Manitoba. Nine Circles Community Health Centre, The Manitoba Harm Reduction Network, Main Street Project and Meeting the Moment are all key

organisations and initiatives supporting underserved groups and guiding this research process.

After initial consultations, we assembled a RAC (online supplemental material 1), which includes an Indigenous elder and Indigenous leaders; community members with lived experience in substance use, houselessness and intergenerational trauma; and academic and scientific researchers of HIV, STBBI and sex-based and gender-based analyses. The RAC guides the research team in research design and community-based engagement methods, data collection tools, culturally safe approaches to recruitment and data collection, review of preliminary findings and knowledge translation strategies. The RAC's continued involvement ensures that the project is culturally safe for the diverse communities involved, the community's best interests are considered and the results and findings are shared appropriately with stakeholders.

Additionally, we convened a PRT of community members with lived experience (online supplemental material 2). The purpose of the PRT is to involve community members as co-researchers during the design of the recruitment strategy and data collection tools, data collection, data analysis, findings and knowledge translation. Our approach to peer engagement focuses on building reciprocal relationships and sharing skills with peer members for each step in the research process, respectfully and safely. PRT meetings are held 2–3 times per month for 2.5 hours. Peers are paid for their time and provided with transportation, childcare support and skill-building activities.

Indigenous peoples make up 18% of the population of Manitoba (compared with 4.9% of the total population in

Canada),[40] yet are over-represented in new HIV diagnoses in Manitoba.[8] While not all participants will self-identify as Indigenous, we have received guidance from peers, RAC members and experts in community-based research to ensure our project remains culturally safe for all. Thus, a critical part of our research process centres around reciprocity. Reciprocity is a critical component of Indigenous culture that should be applied as an overarching stance rather than a particular interaction.[41] Meaningful collaboration with peers, sharing skills (eg, workshops, training sessions) and co-learning are components of our commitment to reciprocity. The RAC emphasised the need for reciprocity during data collection (with PLHIV) suggesting connecting participants with care (if they consent), offering participants other supports (eg, peer and cultural supports, trauma counselling), honorarium, transportation and childcare support. Beyond the data collection, members of the PRT, community members and leaders of community-based organisations are working with the research team to share the preliminary and final findings, mobilise and advocate for improvements and changes and to develop actionable strategies from the findings. These strategies may include training courses to build capacity on related subjects (eg, prevention and treatment of STBBIs and HIV), community-based dissemination strategies and advocacy strategies for increased resources. Likewise, an Indigenous Cultural Advisor joined the team to provide a culturally safe support to participants, peers and the research team. Even though the Cultural Advisor will not be able to advise on indigenous cultural safety for each distinct Indigenous group, she has extensive experience working in community organisations and can provide referrals to culturally safe resources. The Cultural Advisor also educates the research team about the history of Indigenous People in Canada, Indigenous values, traditions, ways of living and culturally safe research approaches.

### Participants and settings

Inclusion criteria for PLHIV include (group 1):

1. Resident of Manitoba
2. 18 years or older
3. Diagnosed with HIV

Inclusion criteria for service providers (group 2):

1. Resident of Manitoba
2. Provide medical, social or other services to PLHIV

The target sample size for this prospective study will be 15–25 PLHIV and 20–30 service providers who work with PLHIV.

### Recruitment process

Purposive sampling will be used due to the distinct eligibility requirements of the participant groups.

### Group 1 recruitment process

Service providers from the Manitoba HIV Program locations (Health Sciences Centre HIV, Nine Circles Community Health Centre and 7th Street Health Access) will

be asked to support the recruitment of PLHIV. Service providers in these locations provide HIV-related support services and have established and trusted relationships with PLHIV. Recruitment materials will be shared with clinicians and service providers to discuss the study with potential participants. If participants are interested, they can contact the research team to discuss the study in more detail and confirm participation.

To support recruiting participants not linked to care, the research team will engage with community organisations supporting PLHIV who may use substances and may be experiencing unstable housing and mental health challenges. There has been a history of poor research relationships with PLHIV that have created experiences of mistreatment, exploitation and disconnect between researchers and communities.[42] Therefore, spending time in community locations potential participants frequent is critical to enable meetings, gain trust and build rapport. Site observations at community organisations will create opportunities for connections and facilitate informal conversations with potential participants about the research. In line with the latest Manitoba HIV Program Report,[8] the purposive sample in group 1 will ensure that at least half of the participants self-identify as women, followed by men and non-binary persons. Similarly, we will aim to include the experiences of Indigenous, PWID and those with unstable housing experiences to accurately represent the people newly diagnosed in Manitoba.[8]

### Group 2 recruitment process

A list of organisations that support and provide services to PLHIV will be compiled to recruit service providers. The RAC and community partners will also be asked to suggest key informants. As with group 1, a purposive sample will be used to identify a representative sample of participants who provide various services and identify as women, men or non-binary. Likewise, the perspectives of black, Indigenous and people of colour service providers will be considered for sampling.

### Data collection

We will collect quantitative and qualitative data focused on understanding HIV care and harm reduction services, social, structural and programmatic factors (table 1).

The research team will greet potential participants who contact the research team by email or phone. Research members will follow a phone or email script to discuss inclusion, scheduling, accessibility, psychological and emotional support for the data collection process. Psychological or emotional support available for this study will include mental health counselling, cultural counselling and peer support. Resource(s) will only be arranged and offered on request, to ensure participant confidentiality. Participants will be provided information and access to healthcare related to HIV, other STBBIs, counselling or primary health through referrals to community organisations. All the field research teams have training in

**Table 1** Data collection, sources, tools and variables collected in this study

| Source | Data type | Tool used | Variables |
|---|---|---|---|
| People living with HIV (PLHIV) | Qualitative data | Interview guide (online supplemental material 3) | Personal description (Q1–3), experience during HIV diagnoses (Q4), services used for HIV care (Q5), barriers and facilitators to HIV care (Q6–8, Q11), impacts of COVID-19 (Q9), safety in healthcare settings (Q10), changes to HIV care (Q12), substance use patterns, types of substances and drug injection practices (Q13–19), harm reduction services (Q20, Q22), substance use and violence (Q21), substance use and HIV care (Q23), COVID-19 and substance use (Q24), best ways to learn about prevention and treatment of STBBIs and harm reduction practices (Q25). |
| | Quantitative data | Sociodemographic and life circumstances survey (online supplemental material 4) | Age (Q1–2), sex at birth, gender identity, sexual orientation, cultural background, marital status, language, religion, education level, place of residence, disabilities, income and housing (Q13–19), experiences with the criminal justice system, including discrimination and stigma because of HIV diagnosis (Q21–28), sexual and testing practices (Q29–36), substance use patterns, quantity, types of substances and drug injection practices (Q37–43), experiences with violence including supports received for healing and involuntary disclosing of HIV status (Q44–48). |
| | | Childhood Trauma Questionnaire[43] | Emotional abuse, physical abuse, sexual abuse, emotional neglect and physical neglect. |
| | | Empower-Making Decisions Survey[44] | Self-efficacy, perceived power, optimism about and control over the future and community optimism |
| Service providers | Qualitative data | Interview Guide (online supplemental material 5) | Sex and gender, role, organisation and services provided (Q1–3), facilitators and barriers to HIV care in PLHIV (Q3–4), barriers specifically for PLHIV who use substances (Q5), the impact of COVID-19 on clients, organisation and personal (Q6–9), types of supports provided for clients (Q10–11), understanding of harm reduction (Q12), prevention of sexually transmitted and bloodborne infections (Q13, Q18), creating safe environments (Q14), HIV policies in Manitoba (Q15–17). |

Q, question; STBBI, sexually transmitted and bloodborne infection.

trauma-informed care, harm reduction, mental health first aid and non-violent crisis intervention. This safety process ensures that participants can receive support, as some of the themes explored may elicit emotional responses.

For PLHIV, data will be collected by the research associate using a combination of face-to-face in-depth semi-structured qualitative interviews and three surveys. The qualitative component will use an interview guide with 26 open-ended questions (online supplemental material 3). The interview guide used for this study has been revised several times by the RAC, the PRT and key stakeholders (eg, service providers). Questions focus on experiences with acquiring and receiving an HIV diagnosis, connecting and receiving HIV care, discrimination and stigma, substance use, past trauma and violence, changes in life due to COVID-19 and knowledge and experiences with harm reduction services (table 1). In addition, PLHIV will be invited to suggest changes to HIV care and broader health and social care in Manitoba. During the screening process, participants will be offered a PRT member as a co-interviewer and a cultural advisor accompanying the research associate. PRT members will be trained to conduct interviews and facilitate surveys. Many PRT members already have experience in community-based research, so they will be familiar with the data

collection tools. At the suggestion of the RAC, we will schedule a break after the interviews to offer various activities (eg, meditation and stretching exercises) and food (eg, granola bars and juice boxes) that create a welcoming and comfortable environment for participants.

Three surveys will be conducted after the break, including a Participant Demographic and Life Circumstances Survey (online supplemental material 4), Childhood Trauma Questionnaire (Short-Form)[43] and the Empower-Making Decisions Survey.[44] The research team developed the Participant Demographics and Life Circumstances survey in consultation with key stakeholders and people with lived/living experiences. It enquires about place of residence, race/ethnicity/culture/language, occupation, gender/sex, religion, education, socioeconomic status and social capital (PROGRESS) factors. Employing PROGRESS indicators provides an equity lens in highlighting health disparities.[45] These questions will allow for describing the structural factors influencing participants' health. This survey also includes questions about substance use, harm reduction knowledge and practices, experiences with the criminal justice system, mental health challenges and experiences of violence and trauma services (table 1). The RAC and PRT extensively reviewed this survey to ensure readability and accuracy to capture the various structural factors PLHIV experience in Manitoba. The second questionnaire, the Childhood

Trauma Questionnaire-Short,[43] is a self-reported inventory that employs Likert scale responses to assess different types of childhood mistreatment and understand the role of historical trauma on life circumstances (table 1). Finally, the Empower-Making Decisions Survey[44] (revised shortened version) employs a Likert scale to understand factors related to resilience and decision-making (table 1). These comprehensive measures will provide a holistic description of the inter-related experiences of HIV care, COVID-19, substance use, PROGRESS indicators, mental health factors, trauma, perceived personal resilience and decision-making of PLHIV in Manitoba. All interviews will take place in a private room to ensure participant confidentiality and researchers' safety. Participants will receive $C50 cash for their time at the end of the interview. Any transportation costs and childcare services will be reimbursed. Interviews may be broken into several sessions based on participants' circumstances and preference.

For service providers, semi-structured interviews will involve 19 open-ended questions (online supplemental material 5 and table 1). The interview guide used for service providers has been revised and edited several times by the RAC and key stakeholders. Questions focus on structural and social factors affecting HIV services and the lives of PLHIV. Service providers will also be asked to provide recommendations to improve HIV care. All interviews will occur over Zoom or Microsoft Teams following the University of Manitoba Virtual Platform Guidelines. Confidentiality and security in videoconferencing software will be ensured by only allowing authenticated users to join the meetings and saving meeting recordings in either a University of Manitoba OneDrive or locally to a University of Manitoba device with disk encryption enabled.

The research team will explain the study and obtain consent from all participants before the data collection. If written consent is not possible, research team members will collect verbal consent recorded at the beginning of the interview. Participants will take a copy of the consent form for their records. Sessions are expected to last between 1 to 2.5 hours. Recordings will be encrypted and stored in a password-protected computer. In addition to recordings, research team members may take notes directly on the interview guide or in notebooks. After each interview, research team members will participate in debriefing sessions to increase study rigour.

This study is part of a broader research that was describing the epidemiology of HIV in Manitoba (Manitoba HIV Program Report[8] and manuscripts in preparation). We collected from clinical charts sociodemographic and clinical information from 517 people newly diagnosed with HIV between 2018 and 2021. Data included >90 variables about age, sex, gender, race/ethnicity, rural/urban, substance use (drugs and route), other infectious diseases and non-communicable diseases/conditions and treatments, primary care information, HIV-related and COVID-19-related information and follow-ups.

## Data analysis

All qualitative and quantitative data will be disaggregated by sex and gender.

All qualitative data will be transcribed verbatim using Otter.ai and checked by the research team to ensure accuracy. Transcripts will be entered into NVivo (V.12) and analysed using grounded theory methods and sex-based and gender-based analysis. Grounded theory provides an iterative framework in which data are collected and analysed simultaneously by immersing the research members in the data through open coding, code analysis and further exploration of new themes.[46] Three independent research team members with experience in qualitative analysis will conduct exploratory coding on an initial subset of the transcripts to establish a codebook for each group. Two independent coders will then establish intercoder reliability to increase rigour. Emerging codes will be grouped into larger categories, and themes will be developed to describe the data within a larger framework. During data analysis, peer debriefing will be used with PRT members to share views (eg, processes, participant engagement) and ideas about codes and themes since this process reduces researcher bias.[47] Negative case analysis will also reduce researcher bias throughout the data analysis process. Finally, an audit trail of decisions through the data collection and analysis will be kept, increasing reproducibility.

Quantitative data from the three questionnaires will be reported using descriptive statistics. These surveys will complement qualitative data by providing information regarding social and structural inequity factors and provide a broader perspective of the experiences of PLHIV in Manitoba.

Following the syndemic model,[33 34] the data collected from clinical charts will provide information about concomitant conditions and how they interact. We will analyse: demographic characteristics: sex at birth, gender, sexual orientation, race/ethnicity and age. Living circumstances: houselessness, rural/urban area. Self-reported main HIV exposure factors: heterosexual sex, gay, bisexual and other men who have sex with men, injection drug/needle use, perinatal acquisition, recipient of blood/blood products, other (which one) and unknown/no identified risk. Clinical diseases and other conditions: STBBI, other comorbidity/condition reported at HIV entry into care, including other infectious diseases non-STBBI, other chronic condition, mental health issues. Disease-disease and social condition-disease interactions include: houselessness (yes/no), injection drug use (yes/no), STBBI at entry into HIV care (yes/no), sex (female/male, there was no intersex reported) and gender (women, men and non-binary), coexisting mental health condition (yes/no). Main outcomes: linkage to HIV care (linked/non-linked) defined as the person who attends an appointment with an HIV clinician within 3 months of HIV diagnosis, or the person started on antiretroviral treatment within 3 months of HIV diagnosis, retention in HIV care (yes/no) defined as the person interacting with

the HIV care during the follow-up and undetectable viral load (yes/no) defined as the person has the viral load <200 copies/mL. With this analysis we want to: (1) uncover whether females and women report distinct combinations of syndemic conditions, (2) examine associations between demographic and living status and syndemic class, (3) assess between-group differences in linkage to HIV care, retention in HIV care and undetectable viral load and (4) test syndemic conditions as a predictor of non-linkage to HIV care, no retention in HIV care and detectable viral load. We will perform a latent class analysis, followed by a logistic regression if the outcomes are >10%, or Poisson regression if the outcomes are <10%.

The quantitative and qualitative data collected prospectively and reported in this study will provide information about how people experience their conditions (social, structural and cultural factors unique to each person), and how HIV care and harm reduction services connect to people.

## ETHICS AND DISSEMINATION

This project received ethics approval from the University of Manitoba Health Ethics Research Board (HS25572; H2022:218), First Nations Health and Social Secretariat of Manitoba, Nine Circles Community Health Centre, Shared Health Manitoba (SH2022:194) and 7th Street Health Access Centre. As some participants will self-identify as First Nations, Métis or Indigenous, the project will follow Indigenous data sovereignty principles. Among these are the First Nation principles of OCAP, Manitoba Métis principles of OCAS, the Principles of Ethical Métis Research and CARE Principles for Indigenous Data Governance, under the leadership of the RAC.

Findings will be disseminated through community-led knowledge translation strategies identified by participants, peers, community members and community organisations; conference presentations, peer-reviewed journal articles, social media, workshops and a dedicated webpage (www.alltogether4ideas.org).

## DISCUSSION

Staggering inequities affect the lives of many groups across Canada.[48] To understand health inequities in Manitoba, it is crucial to contextualise the structural conditions that shape health outcomes. This contextualisation would benefit from using social determinants of health and a syndemic framework in which health crises such as HIV, COVID-19 and substance use interact with each other and with inequitable social conditions to exacerbate adverse health outcomes in underserved groups.[33–35] In Manitoba, PWID, Indigenous peoples and people experiencing houselessness are over-represented in new HIV diagnoses.[8] Similarly, the onset of the COVID-19 pandemic accentuated pre-existing socioeconomic and structural factors, mental health concerns, violence, stigma and discrimination, increased sex and gender disparities and reduced

the resources for STBBI and harm reduction services.[29 31] Complementing qualitative and quantitative data collection with PLHIV and service providers can provide an in-depth understanding of barriers and facilitators to HIV care in Manitoba. Using in-depth semi-structured interviews and questionnaires will enable a broader approach to understanding and addressing Manitoba's various health crises and social inequities. This innovative study uses the WHO Conceptual Framework for Action on the Social Determinants of Health and syndemic frameworks to understand the inter-related experiences of HIV care, COVID-19, substance use and harm reduction, trauma, violence and other social and structural factors.

This project is driven by participatory engagement processes that centre on the knowledge and expertise of community members, organisations and people with lived and living experiences to improve the lives, health and well-being of PLHIV in Manitoba and beyond. Our sample is restricted to people living in metropolitan areas that account for approximately 80% of PLHIV (ie, Winnipeg and Brandon) in Manitoba and individuals who can dedicate the time necessary for the data collection. However, our purposive recruitment strategy to interview people with various backgrounds will ensure we have a representative sample and gain a more comprehensive understanding of the factors that affect the experiences of PLHIV.

Our project will shed light on the growing segment of people missing from the UNAIDS' and Canada's 95-95-95 goal.[9 10] Thus, these results will help to advocate for policy and systemic changes to improve the health and well-being of PLHIV in Manitoba.

**Author affiliations**
[1]Department of Medical Microbiology and Infectious Diseases, Max Rady College of Medicine, Rady Faculty of Health Sciences, University of Manitoba, Winnipeg, Manitoba, Canada
[2]National Collaborating Centre for Infectious Diseases, University of Manitoba, Winnipeg, Manitoba, Canada
[3]Department of Community Health Sciences, Rady Faculty of Health Sciences, University of Manitoba, Winnipeg, Manitoba, Canada
[4]Department of Internal Medicine, Max Rady College of Medicine, Rady Faculty of Health Sciences, University of Manitoba, Winnipeg, Manitoba, Canada
[5]Criminal Justice, The University of Winnipeg, Winnipeg, Manitoba, Canada
[6]Department of Medicine, The University of British Columbia, Vancouver, British Columbia, Canada
[7]Centre for Gender & Sexual Health Equity, The University of British Columbia, Vancouver, British Columbia, Canada
[8]Manitoba HIV Program, Winnipeg, Manitoba, Canada
[9]Nine Circles Community Health Centre, Winnipeg, Manitoba, Canada
[10]Cadham Provincial Laboratory, Shared Health, Winnipeg, Manitoba, Canada
[11]Division of Infectious Disease, The University of British Columbia, Vancouver, British Columbia, Canada
[12]Indigenous Development, University of Winnipeg, Winnipeg, Manitoba, Canada
[13]Laboratory Integration, Office of Population and Public Health, Indigenous Services Canada, Winnipeg, Manitoba, Canada

**Acknowledgements** We are mainly thankful of Peer Research Team members Clara Dan, Joel Baliddawa, Lisa Patrick, Nikki Daniels, Robert Russell, Rebecca Murdock, Freda Woodhouse, Ann Favel, Nadia Mancheese and Marj Schenkels for their invaluable insights. We are thankful to members of the Research Advisory

committee including Dr Jacqueline Gahagan, Elder Margaret Lavallee, Veda Koncan, Margaret Bryan and Jody Jollimore. We are thankful to Heather Pashe (Indigenous Cultural Advisor), some members of Sisters of Fire, Manitoba Harm Reduction Network, Main Street Project, Meeting the Moment, Nine Circles Community Health Centre, Street Connections, 7th Street Health Access Centre and Canadian AIDS Treatment Information Exchange (CATIE) for their ongoing guidance in this project. We are appreciative of all the social workers, nurses, physicians, health education facilitators and service providers in making this study possible. We are thankful to Mariana Echeverri, communications coordinator of this project, who has designed the website, recruitment materials and all communicational pieces regarding this research. Lastly, thanks to the participating institutions: Shared Health, Health Sciences Centre, Nine Circles Community Health Centre Winnipeg, and 7th Street Health Access Centre.

**Contributors** The study concept and design were conceived by ZVR, MH-B and YK with input from CS, EV, LL, KM, KD, JS, KT, LMK, LI, KK, MP, JB, AK, NP, TM and AM. EV will schedule, collect consent, interview, and complete surveys with participants. CS will schedule, collect consent, interview and complete surveys with service providers. EV, CS and ZVR prepared the first draft of the protocol manuscript. LL, KM, KD, JS, KT, LMK, LI, KK, MP, JB, AK, NP, TM, AM and YK, members of the Research Advisory Committee and the Peer Research Team, reviewed, edited and revised it critically for important intellectual content.

**Funding** This work was supported by the Canadian Institute for Health Research (grant number EG5-179453), The Manitoba Medical Service Foundation (grant number: 2021-13), The James Farley Memorial Fund. This research was also supported, in part, by the Canada Research Chairs Programme for ZVR (Award # 950-232963). Production of this manuscript has been made possible in part through a financial contribution of the Public Health Agency of Canada to the National Collaborating Centre for Infectious Diseases.

**Disclaimer** The views expressed here do not necessarily represent the views of the Agency.

**Competing interests** None declared.

**Patient and public involvement** Patients and/or the public were involved in the design, or conduct, or reporting, or dissemination plans of this research. Refer to the 'Methods' section for further details.

**Patient consent for publication** Not applicable.

**Provenance and peer review** Not commissioned; externally peer reviewed.

**Data availability statement** No data are available. Data for this study is not available yet since this is a protocol paper. The study interview guides and questionnaires are available in the supplementary materials section.

**ORCID iD**
Zulma Vanessa Rueda http://orcid.org/0000-0001-6342-1812

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
