## [Reviewer comments · BMJ Open]

ARTICLE DETAILS

TITLE (PROVISIONAL)	Social and structural barriers and facilitators to HIV health care and harm reduction services for people experiencing syndemics in Manitoba: Study protocol
AUTHORS	Rueda, Zulma; Haworth-Brockman, Margaret; Sobie, Cheryl; Villacis, Enrique; Larcombe, Linda; Maier, Katharina; Deering, Kathleen; Sanguins, Julianne; Templeton, Kimberly; MacKenzie, Lauren; Ireland, Laurie; Kasper, Ken; Payne, Michael; Bullard, Jared; Krusi, Andrea; Pick, Neora; Myran, Tara; Meyers, Adrienne; Keynan, Yoav

VERSION 1 – REVIEW

REVIEWER	Singer, Merrill University of Connecticut
REVIEW RETURNED	15-Sep-2022

GENERAL COMMENTS	This paper presents the development of and plans for a study focused on service providers of gendered syndemics of HIV in the context of the COVID-19 epidemic in Manitoba. The study is thoughtfully and sensitively designed and reflects knowledge gained in 30 years of participatory community-based research. The methodology of the study is exemplary. This paper merits publication as a case example of well-planned participatory community-based research that is sensitive to gender and cultural identity issues and has a strong foundation for community involvement, input, and overview. Of concern, however, is the labeling of the study as a syndemics research initiative. Authors provide on the slimmest of definitions or discussion of syndemics. How will they determine if there is a syndemic? It is unclear how the syndemic is to be studied? Assessing the syndemic is not even one of the study's expected outcomes.
--

REVIEWER	Sharma, Bonita B The University of Texas at San Antonio, Social Work
REVIEW RETURNED	13-Feb-2023

GENERAL COMMENTS	Abstract: The goal of the project is clearly defined. Even though this is part of a bigger study, some concepts need to be clearly laid out. Here are a few of my questions: You speak of harm reduction in the goal; how is the concept being applied during the study and measured in the study objectives/outcomes?
--

	In the abstract section, it is not clear how syndemic is being considered. Clearly define what it is and state the context of the syndemic for this study. How many are men and non-binary persons? Indigenous and PWID are not mentioned in the abstract. The demographic in the literature are quite scattered. There's a focus on Indigenous groups, but they represent only 18% of the population (in general). What about the others? Is this study mostly focused on the Indigenous groups? Methods and analysis: The analysis section needs to be more detailed. If the intersectional analysis is being considered, what are the identifiers being evaluated? (The reviewer had to figure it out in the later section). Open-ended interviews and surveys: what is the broad research question for the study and subsequent measures? Page 11: specify RAC members Your primary participant inclusion criteria are PLHIV. Not clear in the outcomes how PWID outcomes are being measured. Ethical Considerations The reciprocity concept is much appreciated. However, it is not sufficient that you provided these temporary gifts/resources during the recruitment and data collection process. Since it is a participatory action research, be specific on how you plan to follow through on the dissemination of research to the research participants and how will they be empowered to take action based on the findings. Patient and Public involvement How will you measure the colonial impact on increased substance use? How will you distinguish its impact on the current generation? For example, are you measuring secondary trauma? If so, how? Data collection Do you plan to use password protection during Zoom interviews? Participant Interview Question Guide: Having Elder support for cultural support needs to be carefully considered since this oust the HIV status of the participant. Please provide more detail on how the confidentiality will be upheld during and long after the research.
--	--

VERSION 1 – AUTHOR RESPONSE

Comments from the Reviewer 1:

1. This paper presents the development of and plans for a study focused on service providers of gendered syndemics of HIV in the context of the COVID-19 epidemic in Manitoba. The study is thoughtfully and sensitively designed and reflects knowledge gained in 30 years of participatory community-based research. The methodology of the study is exemplary. This paper merits publication as a case example of well-planned participatory community-based research that is sensitive to gender and cultural identity issues and has a strong foundation for community involvement, input, and

overview. Of concern, however, is the labeling of the study as a syndemics research initiative. Authors provide on the slimmest of definitions or discussion of syndemics. How will they determine if there is a syndemic? It is unclear how the syndemic is to be studied? Assessing the syndemic is not even one of the study's expected outcomes.

a. Answer: We thank the reviewer for their kind comments regarding. We have tried to incorporate knowledge gained from many stakeholders to ensure this study is grounded in the experiences of people living with HIV and frontline workers, and prompts actionable changes at a community level.

b. We have changed the abstract, the introduction, included in the methods section a "conceptual framework", added Figure 1 and Table 1, and expanded the data collection and analysis sub-sections. In summary, with all the changes in the revised manuscript, we tried to ensure a thorough description of our conceptual frameworks: World Health Organization Conceptual Framework for Action on the Social Determinants of Health and the syndemic framework. In particular, we have changed the abstract and included the co-occurring pandemics and structural inequities that are interacting and perpetuating health crises in underserved groups in Manitoba. Additionally, we have reformatted the introduction so that the initial paragraphs describe the health crises, social inequities, and social determinants of health that are occurring in Manitoba. We added a new section titled "Conceptual Framework" at the start of the Methods and Analysis section to describe these two frameworks and how they will be considered for this project.

c. As to how the syndemic will be studied, we have created a table (table 1) that highlights the specific questions and themes that will be asked during data collection. It has also been clarified in the data collection section (Interview guide section in particular) the themes that participants will be asked about during the interviews will provide a picture of their experiences. We believe that these themes (substance use, trauma, violence) and questions about barriers and facilitators to HIV care and harm reduction services, and the effects of Covid-19 will provide valuable qualitative data to determine if there is a syndemic, and its nature. Additionally, quantitative data will provide descriptive statistics of the social and structural inequity indicators that were collected through the surveys. These descriptive statistics will complement the qualitative data by providing a broader framework of understanding of the experiences of PLHIV in Manitoba.

In addition, as we mentioned in the objectives, this study is part of a broader research that was studying the epidemiology of HIV in Manitoba (Manitoba HIV Program Report [1] and manuscripts in preparation). We collected data from clinical charts sociodemographic and clinical information from 517 people newly diagnosed with HIV between 2018-2021. Data included >90 variables about age, sex, gender, race/ethnicity, rural/urban, substance use (drugs and route), other infectious diseases and non-communicable diseases/conditions and treatments, primary care information, HIV- and Covid-19 related information, and follow-ups. Following the syndemic model [2], the data collected from clinical charts will provide information about concomitant conditions and how they interact. The quantitative and qualitative data collected prospectively and reported in this paper, will provide information about how people experience their conditions (social, structural and cultural factors), and how HIV care and harm reduction services connect to people.

d. The study outcomes section was modified to reflect that the findings will be plotted within a World Health Organization Conceptual Framework for Action on the Social Determinants of Health and syndemic framework to understand the interconnectedness of the health crises.

Comments from the Reviewer 2:

1. Abstract: The goal of the project is clearly defined. Even though this is part of a bigger study, some concepts need to be clearly laid out. Here are a few of my questions

a. Answer: Thank for your comment. We have clarified our manuscript according to your comments.

2. You speak of harm reduction in the goal; how is the concept being applied during the study and measured in the study objectives/outcomes?

a. Answer: We have provided a specific description of harm reduction in the 5th paragraph of the introduction. Additionally, we have clarified in the data collection section the specific variables we will address with the qualitative and quantitative methods (please see table 1). Participants will be asked about their knowledge and frequency of substance use during the qualitative interview. Additionally, they will be asked about frequency and types of harm reduction strategies used during the quantitative demographic survey.

3. In the abstract section, it is not clear how syndemic is being considered. Clearly define what it is and state the context of the syndemic for this study.

a. Answer: The abstract has been revised to reflect a clearer problem. We also include in the methods a summary of the information that we will collect. The introduction was reformatted to better describe the syndemic context. In particular, the various health crises and its impacts are described first. Then, we described the known social inequities affecting underserved groups in Manitoba. We finish the introduction by consolidating this context. We also included a sub-section in the methods about our conceptual framework, added Figure 1 to situate our research, and added Table 1 to summarize the data collection, tools, sources and themes.

4. How many are men and non-binary persons? Indigenous and PWID are not mentioned in the abstract. The demographic in the literature are quite scattered. There's a focus on Indigenous groups, but they represent only 18% of the population (in general). What about the others? Is this study mostly focused on the Indigenous groups?

a. Answer: Thanks for these questions. Despite the fact that Indigenous people represent 18% of the population, among people newly diagnosed with HIV, more than 70% self-identified as Indigenous people, and in 2021, more than 80%. Our group did a study to identify the epidemiology of HIV in Manitoba between 2018 and 2021. The resulting Manitoba HIV Program Report was released on December 1, 2022 (available at: www.mbhiv.ca)

We found that 50% are females and 50% males. Almost all people self-identified as women and men. Among 517 people newly diagnosed with HIV, 5 self-identified as transwomen or transmen, and 3 self-identified as non-binary persons. Regarding sexual orientation, more than 80% self-identify as heterosexual, 12.4% self-reported as gay and 4.7% as bisexual. In the report, we used sex at birth to disaggregate the information because there was no missing data in this variable. We found that 48% of females and 1/3 of males experienced houselessness, and among people who use drugs, 70.7% of females and 35.1% of males inject drugs, the most used injection drug was methamphetamine (78.8% of females and 59.7% of males). These results are striking and definitely are different from the rest of Canada. This is the reason we think the research we are reporting in this protocol is very relevant.

b. We have added new reference from the Manitoba HIV Program Report in the first paragraph of the introduction stating that PWID and Indigenous peoples are overrepresented in the new diagnoses in the province. So, we have stated that we acknowledge that not all participants will be Indigenous, yet we have placed a great focus on Indigenous groups since there is an overrepresentation in new diagnoses.

c. We have added a section under the last paragraph of the Group 1: Recruitment Process. In this section, we have highlighted that our sample aims to be representative of those newly diagnosed in Manitoba, such that half of the sample will self-identify as women, followed by men and those who identify as non-binary. In this section, we have added that in line with the Manitoba HIV report of new diagnoses, we will look out for people who self-identify as Indigenous, PWID, or those with experiences of houselessness to ensure sample representation.

5. Methods and analysis: The analysis section needs to be more detailed.

a. Answer: The data analysis section has been expanded to include specific processes that will occur for the qualitative data analysis. In particular, we have expanded how we will use grounded theory to develop initial codebooks, create intercoder reliability, and group emerging codes into categories and

themes. Quantitative data will be used to provide descriptive statistics of the social and structural indicators of inequity collected through the surveys. These descriptive statistics will complement the qualitative data, together providing a broader framework of understanding of the experiences of PLHIV in Manitoba. We also added the paragraph that we pasted on Reviewer's comment 1.

6. If the intersectional analysis is being considered, what are the identifiers being evaluated? (The reviewer had to figure it out in the later section).

a. Answer: The intersectional identifiers of interest have been added earlier in the methods, in the conceptual framework and Figure 1, and the Table 1, as well as, in the analysis section.

7. Open-ended interviews and surveys: what is the broad research question for the study and subsequent measures? Page 11: specify RAC members

a. Answer: We included the broader question at the end of the introduction. We also included the outcomes and Table 1.

b. Regarding the RAC members, we summarized the composition within the text. Due to the word limit within the main article, supplementary material 2 details the terms of references of the Research Advisory Committee and the names of each member with their affiliations.

8. Your primary participant inclusion criteria are PLHIV. Not clear in the outcomes how PWID outcomes are being measured.

a. Answer: All participants are people living with HIV. Among them, there are people who inject drugs and people who do not. Based on the Manitoba HIV Program report, we found that more than half of people newly diagnosed inject drugs. We included in the objective and outcomes access to harm reduction services as the main outcome, and in the data that we are collecting, a more detailed description of our measures in Table 1 and within the text using the quantitative surveys (substance use patterns, quantity, types of substances, and drug injection practices (Q37-43), supplementary material 4), and qualitative interview (substance use patterns, types of substances, and drug injection practices (Q13-19), harm reduction services (Q20, Q22), substance use and violence (Q21), substance use and HIV care (Q23), Covid-19 and substance use (Q24), best ways to learn about prevention and treatment of STBBI's and harm reduction practices (Q25), supplementary material 3).

9. The reciprocity concept is much appreciated. However, it is not sufficient that you provided these temporary gifts/resources during the recruitment and data collection process. Since it is a participatory action research, be specific on how you plan to follow through on the dissemination of research to the research participants and how will they be empowered to take action based on the findings.

a. Answer: We agree and we aim to go beyond honorarium. We have added a paragraph within the community engagement and within the data collection process addressing reciprocity and actionable strategies beyond data collection. Participants will be included (if they consent) in deciding on follow-up dissemination of the results and knowledge translation strategies. We have been working with the peer research team for almost one year to share skills and co-develop capacity building and knowledge translation strategies. For example, with the Manitoba HIV Program Report we shared the results before its publication, over three months and in more than 20 meetings, we met with more than 200 people (community members, people with lived experiences, front-line workers, stakeholders, policy and decision-makers, etc). One concrete outcome of our process was the Calls to Action. Currently, there has been other concrete actions and strategies to mobilize and advocate for changes, led by multiple organizations, including community-based organizations. We are also documenting all community-engagement processes and lessons learnt to submit for publication. On our website (www.alltogether4ideas.org) we report weekly on all activities that we do regarding this process.

10. How will you measure the colonial impact on increased substance use? How will you distinguish its impact on the current generation? For example, are you measuring secondary trauma? If so, how?

a. Answer: Thanks for your question. We are not measuring the colonial impact on increased substance use. This specific outcome would require a different kind of study. We wanted to acknowledge the history that creates some of the inequities. However, to avoid misleading messages, we deleted reference to this aspect.

11. Do you plan to use password protection during Zoom interviews?

a. Answer: We have added an elaborated description of the process that ensures confidentiality, privacy and security in online data collection. We have made reference to the specific University of Manitoba Virtual Platform Guidelines that were used in our Standard Operating Procedures (SOPs). Only authenticated persons can join the zoom; all interviews are stored password protected in computers within the University of Manitoba, which also requires a password.

12. Participant Interview Question Guide: Having Elder support for cultural support needs to be carefully considered since this oust the HIV status of the participant. Please provide more detail on how the confidentiality will be upheld during and long after the research.

a. Answer: Thanks for this question. To address concerns of confidentiality (outing participants) we changed in the data collection section. Now we have clarified that any additional supports or services during or after data collection will only be arranged at the request of the participant.

VERSION 2 – REVIEW

REVIEWER	Singer, Merrill University of Connecticut
REVIEW RETURNED	17-Apr-2023

GENERAL COMMENTS	This remains an interesting and potentially important study will rigorous attention to doing useful, grounded, ethical, and well guided community-based research. However, several problems remain. A problem still exists with the conceptualization of syndemics. The first sentence of of the revised Conceptual Framework mis-defines syndemics as 2 or more diseases converging (but the real point is actual interaction, which is not clear from “converging”) and social and structural inequalities as a different thing which the authors will bring together. The whole point of syndemics was to bring interacting diseases and inequalities together; that is the very definition of syndemics. This is fixed in the 2nd paragraph of the section, but it is not clear why the original separation is made. Then the problem is compounded again by saying these two theoretical components will guide the study when, as noted, only one theoretical component-syndemics-encompasses both interacting diseases and intersecting social and structural inequalities. This conceptual confusion is repeated again in the first line of the data collection section, splitting syndemics and social/structural factors into 2 different things. For purposes of data collection, two strategies are needed: one focused on diseases present in a population, and one focused on social inequalities. But a syndemics framework treats these as two steps in the same dance, culminating in the finale: disease-disease/inequality interaction. A problem still exists in the data analysis section which states: the data collected from clinical charts will provide information about concomitant conditions and following the syndemic model how they interact. Need to explain how disease interactions will be recorded in or extractable from the Charts. Just how will disease interaction be identified and analyzed. This is far from clear in a
--

	one sentence statement which implies this task presents no challenges.
--	--

REVIEWER	Sharma, Bonita B The University of Texas at San Antonio, Social Work
REVIEW RETURNED	30-Apr-2023

GENERAL COMMENTS	The authors have addressed all of the issues I highlighted satisfactorily. In my opinion, there is a couple of minor additions that are needed: Abstract: Add a line on the relevance of the study's problem given the decline in social distancing and other COVID-19-related restrictive measures. Intro: Page 8: Add an evidence-based sentence or two on how pandemic-related disruption continues to create the barrier to access and why studying the pandemic's effect is relevant.
---

VERSION 2 – AUTHOR RESPONSE

Comments from the Reviewer 1:

1. Dr. Merrill Singer, University of Connecticut Comments to the Author: This remains an interesting and potentially important study will rigorous attention to doing useful, grounded, ethical, and well guided community-based research. However, several problems remain. A problem still exists with the conceptualization of syndemics. The first sentence of of the revised Conceptual Framework mis-defines syndemics as 2 or more diseases converging (but the real point is actual interaction, which is not clear from “converging”) and social and structural inequalities as a different thing which the authors will bring together. The whole point of syndemics was to bring interacting diseases and inequalities together; that is the very definition of syndemics. This is fixed in the 2nd paragraph of the section, but it is not clear why the original separation is made. Then the problem is compounded again by saying these two theoretical components will guide the study when, as noted, only one theoretical component-syndemics-encompasses both interacting diseases and intersecting social and structural inequalities. This conceptual confusion is repeated again in the first line of the data collection section, splitting syndemics and social/structural factors into 2 different things. For purposes of data collection, two strategies are needed: one focused on diseases present in a population, and one focused on social inequalities. But a syndemics framework treats these as two steps in the same dance, culminating in the finale: disease-disease/inequality interaction.

Answer: Thanks. You are correct. We used convergence to use our own words, but to avoid misleading definitions, we used the syndemics definition in quotes with the reference.

Regarding the sentence “the social and structural inequalities as a different thing which the authors will bring together”, we did not aim to propose that these are two different theories. What we wanted to highlight from the WHO document was the definition of health as a social construct rather than a biological process.

We reorganized this section starting with the syndemic definition, then highlighting health as a social construct using the definition of the WHO.

The current conceptual framework of the paper reads as follows:

“The syndemic framework proposed by Singer [33] focused on understanding the interconnectedness of mutually reinforcing biomedical crises within their socioeconomic contexts. The syndemic theory “involves the adverse interaction between diseases and health conditions of all types (eg, infections [HIV and STBI], chronic non-communicable diseases, mental health [issues], behavioural conditions, [substance use], toxic exposure, and malnutrition) and are most likely to emerge under conditions of health inequality caused by poverty, stigmatization, stress, [trauma], or structural violence” [33,34].

Imperative to syndemic theory is the recognition that disparate social conditions not only enable health crises but maintain and reinforce them [33]. Social and structural health inequities that disproportionately affect seldom-heard and marginalized populations in Manitoba require understanding of health as a social construct, rather than a biological process as defined by the World Health Organization Conceptual Framework for Action on the Social Determinants of Health (CSDH) [35]. The framework describes how social, economic, and political values and policies create a socioeconomic position, for people in societies, influencing opportunities for income, education, depending on race/ethnicity, gender, and other factors [35]. These structural determinants of health inequities interact with intermediary determinants of health such as material circumstances, and psychosocial, biological (including sex) and behavioural factors [35]. Using health as a social construct helps to push understanding of inequities beyond a focus on individual health and behaviors, framing health as a social construct dependent on the interplay of numerous interacting systems and circumstances.

This syndemic theory is particularly valuable in the context of communicable diseases and seldom-heard and marginalized peoples' health, as it brings interacting diseases and inequalities together, and it emphasizes the political, social, historical, and economic factors that have negatively affected their health outcomes [34,36]. Figure 1 summarizes our approach for this study.”

Regarding data collection, we split the variables because, in the previous round of reviews, one of the reviewers requested that we explicitly state which social and structural determinant variables we will collect. We deleted the sentence that made it appear as if we are splitting syndemics and social and structural factors, and just mention which variables we will collect. The first section is focused on the social inequalities and the experiences faced by people living with HIV and services providers, and the last part of the data collection is focused on the diseases present in people newly diagnosed with HIV.

With all of these changes, we addressed Dr. Singer's comments.

2. A problem still exists in the data analysis section which states: the data collected from clinical charts will provide information about concomitant conditions and following the syndemic model how they interact. Need to explain how disease interactions will be recorded in or extractable from the Charts. Just how will disease interaction be identified and analyzed. This is far from clear in a one sentence statement which implies this task presents no challenges.

Answer: Thanks. The reason we do not expand on this is because the retrospective piece of all 517 people newly diagnosed with HIV between 2018 and 2021 was done and completed while the paper was under review. The BMJ Open journal clearly states in the instructions to authors that protocol papers cannot report research that is completed. For that reason, we stated since we submitted the paper that “This study is part of a broader research that was describing the epidemiology of HIV in Manitoba (Manitoba HIV Program Report [8] and manuscripts in preparation).”, and briefly summarized all variables that we collected in the data collection section: “We collected from clinical

charts sociodemographic and clinical information from 517 people newly diagnosed with HIV between 2018-2021. Data included >90 variables about age, sex, gender, race/ethnicity, rural/urban, substance use (drugs and route), other infectious diseases and non-communicable diseases/conditions and treatments, primary care information, HIV- and Covid-19 related information, and follow-ups.”

In the data analysis section, we expanded Dr. Singer’s request as follows:

“Following the syndemic model [33,34], the data collected from clinical charts will provide information about concomitant conditions and how they interact. We will analyze: Demographic characteristics: sex at birth, gender, sexual orientation, race/ethnicity, and age. Living circumstances: houselessness, rural/urban area. Self-reported main HIV exposure factors: heterosexual sex, gay, bisexual, and other men who have sex with men, injection drug/needle use, perinatal acquisition, recipient of blood/blood products, other (which one), and unknown/no identified risk. Clinical diseases and other conditions: STBBI, other comorbidity/condition reported at HIV entry into care, including other infectious diseases non-STBBI, other chronic condition, mental health issues. Disease-disease and social condition-disease interactions include: houselessness (yes/no), injection drug use (yes/no), STBBI at entry into HIV care (yes/no), sex (female/male, there was no intersex reported) and gender (women, men and non-binary), coexisting mental health condition(yes/no). Main outcomes: linkage to HIV care (linked/non/linked) defined as the person who attends an appointment with an HIV clinician within three months of HIV diagnosis, or the person started on antiretroviral treatment within three months of HIV diagnosis, retention in HIV care (yes/no) defined as the person interacting with the HIV care during the follow-up, and undetectable viral load (yes/no) defined as the person has the viral load <200 copies/ml. With this analysis we want to: 1) uncover whether females and women report distinct combinations of syndemic conditions, 2) examine associations between demographic and living status and syndemic class, 3) assess between-group differences in linkage to HIV care, retention in HIV care, and undetectable viral load, and 4) test syndemic conditions as a predictor of non-linkage to HIV care, no retention in HIV care, and detectable viral load. We will perform a latent class analysis, followed by a logistic regression if the outcomes are higher than 10%, or Poisson regression if the outcomes are less than 10%.

The quantitative and qualitative data collected prospectively and reported in this paper, will provide information about how people experience their conditions (social, structural and cultural factors unique to each person), and how HIV care and harm reduction services connect to people.”

Comments from the Reviewer 2:

1. Dr. Bonita B Sharma, The University of Texas at San Antonio Comments to the Author: The authors have addressed all of the issues I highlighted satisfactorily. In my opinion, there is a couple of minor additions that are needed:

a. Answer: Thanks, we appreciate your comment.

2. Abstract: Add a line on the relevance of the study’s problem given the decline in social distancing and other COVID-19-related restrictive measures.

a. Answer: Thanks. The abstract currently has this sentence “The Covid-19 pandemic resulted in increased sex and gender disparities in disease risk and mortalities, decreased harm reduction services and reduced access to health care. These health crises intersect with increased drug use

and drug poisoning deaths, homelessness, and other structural and social factors most acutely among historically underserved groups.”

3. Intro: Page 8: Add an evidence-based sentence or two on how pandemic-related disruption continues to create the barrier to access and why studying the pandemic’s effect is relevant.

Answer: Thanks. The introduction has the following paragraphs regarding COVID-19 pandemic and its effects: “The Covid-19 pandemic has dramatically affected the health of many individuals, especially among people already burdened by social and structural health inequities [24]. The pandemic also exacerbated mental health symptoms in many groups [25]. Data from Manitoba shows that older adolescents and young adults self-reported increased stress/anxiety and depression, alcohol consumption and substance use, and conflict with family members and intimate partner[26], with those self-identifying as women experiencing a higher mental health burden [27,28], financial hardship, and interpersonal conflicts [26]. For HIV care, the reduction or complete closure of HIV treatment centres and services in some jurisdictions placed the cascade of care for PLHIV at risk of breaking down [24]. Among PWID, Covid-19 restrictions limited safe spaces for substance use, disrupted drug supplies, and restricted the availability of medical, community, and traditional resources [29]. Harm reduction encompasses strategies to reduce the health, social, and economic factors that harm PWID and their networks [30]. The Public Health Agency of Canada conducted an online survey among people self-identifying as PWID in the past six months to understand how sexually transmitted and blood borne infections (STBBIs) and harm reduction services changed during Covid-19 [31]. The survey reported 52% of respondents increased their personal use of methamphetamine, and more than half increased their use of alcohol and other drugs [31]. As well, 50% of PLHIV had challenges getting to a provider or clinic since the onset of Covid-19 [31]. More than half of the respondents had difficulty accessing STBBI-related services, needle and syringe distribution programs, and naloxone training [31]. While these results highlight concerning trends, data were not reported by sex and gender, hindering an intersectional analysis. Also, the study's online modality may not have reached PWID who may also be coping with unstable housing, poverty, and other structural factors, likely underestimating the disruption of services for those experiencing the most extreme social inequities. Locally, the province of Manitoba reported record fatalities from toxic drug overdoses [32], and PWID described harm reduction services limitations due to Covid-19 [29]. However, more research is needed to understand specific service limitations and how these limitations interact with other health crises and social inequities.”